# Clinical Lymph Node Involvement as a Predictor for Cancer-Specific Survival in Patients with Penile Squamous Cell Cancer

Makoto Kawase [1], Kimiaki Takagi [2], Kei Kawada [3], Takashi Ishida [4], Masayuki Tomioka [5], Torai Enomoto [6], Shota Fujimoto [7], Tomoki Taniguchi [8], Hiroki Ito [9], Koji Kameyama [10], Toru Yamada [11], Kota Kawase [1], Daiki Kato [1], Manabu Takai [1], Koji Iinuma [1], Keita Nakane [1] and Takuya Koie [1,*]

1 Department of Urology, Gifu University Graduate School of Medicine, Gifu 5011194, Japan; buki2121@gifu-u.ac.jp (M.K.); stnf55@gifu-u.ac.jp (K.K.); andreas7@gifu-u.ac.jp (D.K.); takai_mb@gifu-u.ac.jp (M.T.); kiinuma@gifu-u.ac.jp (K.I.); keitaco@gifu-u.ac.jp (K.N.)
2 Department of Urology, Daiyukai Daiichi Hospital, Ichinomiya 4918551, Japan; kimiaki_takagi5619@yahoo.co.jp
3 Department of Urology, Gifu Prefectural General Medical Center, Gifu 5008717, Japan; keinedvedon@yahoo.co.jp
4 Department of Urology, Gifu Municipal Hospital, Gifu 5008513, Japan; justaskaxis@gmail.com
5 Department of Urology, Japanese Red Cross Takayama Hospital, Takayama 5068550, Japan; tomiokam@gifu-u.ac.jp
6 Department of Urology, Matsunami General Hospital, Hashima-gun 5016062, Japan; try@mghg.jp
7 Department of Urology, Ogaki Municipal Hospital, Ogaki 5038502, Japan; f19533612@gmail.com
8 Department of Urology, Chuno Kosei Hospital, Seki 5013802, Japan; tomokidbx@gmail.com
9 Department of Urology, Toyota Memorial Hospital, Toyota 4718513, Japan; seanoel2@gmail.com
10 Department of Urology, Central Japan International Medical Center, Minokamo 5058510, Japan; i2001029@yahoo.co.jp
11 Department of Urology, Tokai Central Hospital, Kakamihara 5048601, Japan; toru.yamada@tokaihp.jp
* Correspondence: goodwin@gifu-u.ac.jp; Tel.: +81-582306000

**Abstract:** We aimed to identify prognostic predictive factors of patients with penile squamous cell carcinoma (PSCC). This retrospective study reviewed the clinical and pathological data of patients with PSCC at 10 institutions in Japan between January 2008 and December 2019. The primary endpoint was cancer-specific survival (CSS). We also identified useful predictive factors for CSS in patients with PSCC. In total, 64 patients with PSCC were enrolled. At the end of the follow-up period, 15 patients (23.4%) died owing to PSCC and six (9.4%) died owing to other causes. The 2- and 3-year CSS rates were 78.9% and 76.6%, respectively. Using the Kaplan–Meier method, the Eastern Cooperative Oncology Group performance status 0, serum albumin levels $\geq 4.2$ g/dL, hemoglobin levels $\geq 13.2$ g/dL, C-reactive protein levels $< 0.21$ mg/dL, clinical T stage $\leq 2$, clinically negative lymph node (LN) status, and tumor size $< 30$ mm were associated with a significantly better CSS. In the multivariate analysis, the clinically positive LN status was a significant predictive factor for CSS in patients with PSCC. Further prospective large-scale and long-term studies are required to validate our findings.

**Keywords:** penile cancer; cancer-specific survival; lymph node involvement; inflammatory markers

## 1. Introduction

Penile squamous cell carcinoma (PSCC) is a relatively rare disease among genitourinary tumors with approximately 26,000 new cases worldwide annually [1]. Although the overall incidence for PSCC in North America and Europe is <1 per 100,000 men, it is a much more common disease in some parts of Asia, Africa, and South America, with an estimated incidence of up to 50 per 100,000 men [2,3]. Especially in Japan, the age-adjusted incidence of PSCC was reportedly 0.2 per 100,000 men in a population-based study of PC from 15 Japanese selected cancer registries [4]. The etiology of PSCC is multifactorial with

risk factors such as smoking, chronic inflammation, balanitis, history of phimosis, poor hygiene, socioeconomic status, and human papilloma virus (HPV) infection, particularly types 16 and 18 [3,5,6]. Lymph node metastasis (LNM) at initial diagnosis is the most important prognostic marker [7,8]. Therefore, the vital steps for developing treatment strategies in patients with PSCC are the early detection of LNM, appropriate surgical treatment, and neoadjuvant or adjuvant systemic chemotherapy [1,9,10]. However, recent studies have evaluated the prognostic role of serum inflammatory markers, including C-reactive protein (CRP), serum albumin, neutrophil–lymphocyte ratio (NLR), platelet–lymphocyte ratio (PLR), and lymphocyte–monocyte ratio (LMR), because of the association between cancer development and prognosis and systemic inflammation for many urologic malignancies [9,11–14]. However, it is difficult to precisely predict oncological outcomes such as cancer-specific survival (CSS) in patients with PSCC with versus without LNM at the initial diagnosis using inflammatory biomarkers or clinical covariates. Thus, this study aimed to identify prognostic predictive factors of patients with PSCC.

## 2. Materials and Methods

### 2.1. Patient Population

We reviewed clinical and pathological data of patients with penile cancer (PC) at 10 institutions in Japan between January 2008 and December 2019. The inclusion criteria were PC diagnosed as squamous cell carcinoma (SCC) in biopsy or surgical specimens. Patients diagnosed with other penile diseases, including extramammary Paget's disease, Bowen's disease, malignant melanoma, or metastatic PC from other organs, were excluded.

The collected clinicopathological data included age, Eastern Cooperative Oncology Group performance status (ECOG-PS) [15], Alb level, Hb level, CRP level, tumor size, lymph node (LN) involvement, number of LNM, neutrophil count, lymphocyte count, thrombocyte count, NLR, PLR, and SCC antigen before PSCC treatment. Tumor staging was performed according to the staging system defined in the American Joint Committee on Cancer staging manual [16]. Regarding lymph node (LN) involvement, positive lymph node metastasis (LNM) was defined as the minor LN diameter of ≥15 mm before PSCC treatment.

This study was approved by the Institutional Review Board of Gifu University (approval number: 2020-271) and the respective institutional review boards. Patient consent was not required owing to the study's retrospective nature. The provisions of the ethics committee and the ethics guidelines in Japan did not require written consent since the study information was disclosed to the public in case of retrospective and/or observational studies using materials such as existing documentation. The details of the study can be found at http://www.med.gifu-u.ac.jp/file/2020-271.pdf (accessed on 21 November 2021).

### 2.2. Endpoints and Statistical Analysis

The primary endpoint of this study was CSS. The JMP 14 software (SAS Institute Inc., Cary, NC, USA) was used for data analysis. The follow-up duration was defined as the interval from the initial diagnosis of PSCC to the last follow-up examination or the documented date of death, whichever occurred first. CSS was estimated using the Kaplan–Meier method. Survival according to the subgroup was analyzed using the log-rank test. Multivariate Cox proportional hazards analysis was performed. The cutoff values for the covariates were defined as the minimum value for $(1 - \text{sensitivity})^2 + (1 - \text{specificity})^2$ according to the area under the receiver operating characteristic curve and CSS and multivariate analyses [17]. A two-sided 5% significance level was used for all statistical inferences.

## 3. Results

### 3.1. Patient Characteristics

Demographic data of the enrolled patients are presented in Table 1. Among patients who underwent surgery, 49 (76.6%) and 7 (10.9%) underwent partial and total penectomies, respectively. Among them, 16 (28.6%) underwent inguinal pelvic lymph node (LN) dissection. Two patients received neoadjuvant chemotherapy and four received adjuvant

chemotherapy. One patient (1.6%) underwent radiation of the origin, bilateral inguinal area, and the whole pelvis owing to poor general condition. Six patients chose best supportive care because of poor ECOG-PS.

**Table 1.** Patient characteristics.

| Variables | |
|---|---|
| Age (year, median, IQR) | 74 (63–82) |
| ECOG Performance Status (number, %) | |
| 0 | 31 (48.4) |
| 1 | 18 (28.1) |
| 2 | 8 (12.5) |
| 3 | 6 (9.4) |
| 4 | 1 (1.6) |
| Clinical T stage (number, %) | |
| Tis | 1 (1.6) |
| T1 | 26 (40.6) |
| T2 | 20 (31.3) |
| T3 | 14 (21.9) |
| T4 | 2 (3.1) |
| Clinical N stage (number, %) | |
| 0 | 38 (59.4) |
| 1 | 9 (14.1) |
| 2 | 8 (12.5) |
| 3 | 9 (14.1) |
| Clinical stage (number, %) | |
| Is | 1 (1.6) |
| 1 | 21 (32.8) |
| 2 | 14 (21.9) |
| 3 | 17 (26.6) |
| 4 | 11 (17.6) |
| Treatment modality (number, %) | |
| Surgery only | 51 (79.7) |
| Surgery and chemotherapy | 6 (9.4) |
| Radiation only | 1 (1.6) |
| Best supportive care | 6 (15.6) |
| Albumin (g/dL, median, IQR) | 4.2 (3.7–4.4) |
| Hemoglobin (g/dL, median, IQR) | 13.2 (11.4–14.8) |
| C-reactive protein (mg/dL, median, IQR) | 0.21 (0.06–1.28) |
| Tumor size (mm, median, IQR) | 30 (20–40) |
| Number of LNM (number, median, IQR) | 2 (1–6) |
| NLR (median, IQR) | 3.05 (2.03–4.96) |
| PLR (median, IQR) | 148 (105–228) |
| Pretreatment SCC (ng/mL, median, IQR) | 1.65 (1.23–4.18) |
| Follow-up period (months, median, IQR) | 26.0 (9.0–61.3) |

IQR: interquartile range; ECOG: The Eastern Cooperative Oncology Group; LNM: lymph node metastasis; NLR: neutrophil-to-lymphocyte ratio; PLR: platelet-to-lymphocyte ratio; SCC: squamous cell carcinoma antigen.

*3.2. Oncological Outcomes*

At the end of the follow-up period, 15 patients (23.4%) died owing to PSCC and six (9.4%) died owing to other causes (details unknown). The 2- and 3-year CSS rates were 78.9% and 76.6%, respectively.

The 2-year CSS rate was 93.0% for patients with an ECOG-PS score of 0 and 63.8% for those with a score of $\geq 1$ ($p = 0.001$). The 2-year CSS rates were 90.3% and 66.0% among patients with serum albumin (Alb) levels of $\geq 4.2$ g/dL and <4.2 g/dL, respectively ($p < 0.012$; Figure 1A); 90.2% and 64.3% among those with hemoglobin (Hb) levels of $\geq 13.2$ g/dL and <13.2 g/dL, respectively ($p = 0.008$; Figure 1B); and 92.9% and 59.7% among those with C-reactive protein (CRP) levels of $\geq 0.21$ mg/dL and <0.21 mg/dL, respectively ($p = 0.004$; Figure 1C).

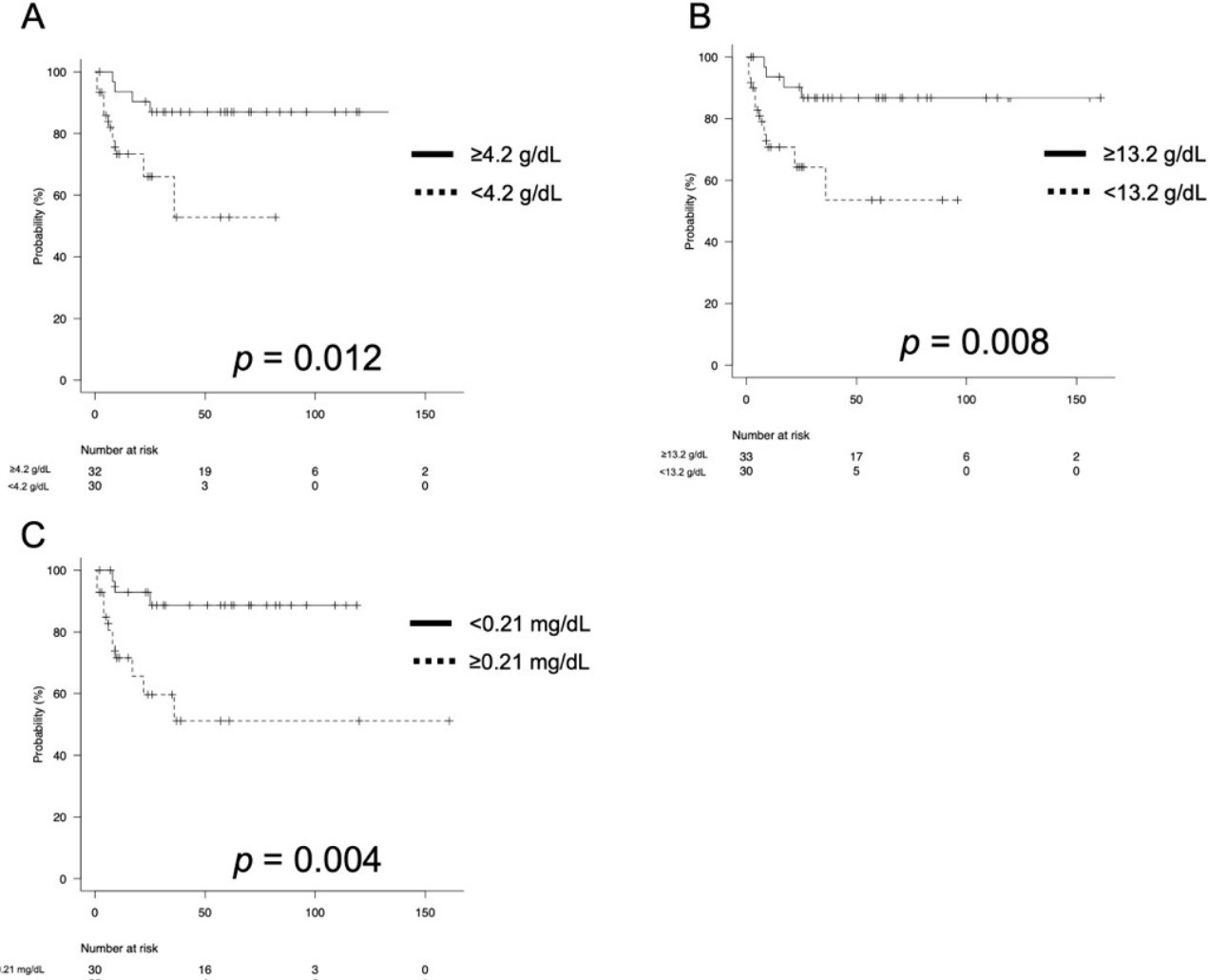

**Figure 1.** The Kaplan–Meier estimates of cancer-specific survival (CSS) according to serum albumin (Alb) levels stratified by a cutoff of 4.2 g/dL (**A**), hemoglobin (Hb) levels stratified by a cutoff of 13.2 g/dL (**B**) and C-reactive protein (CRP) levels stratified by a cutoff of 0.21 mg/dL (**C**). The 2-year CSS rates were 90.3% and 66.0% among patients with Alb levels of ≥4.2 g/dL and <4.2 g/dL, respectively ($p < 0.012$; Figure 1A); 90.2% and 64.3% among those with Hb levels of ≥13.2 g/dL and <13.2 g/dL, respectively ($p = 0.008$; Figure 1B); and 92.9% and 59.7% among those with CRP levels of ≥0.21 mg/dL and <0.21 mg/dL, respectively ($p = 0.004$; Figure 1C).

The 2-year CSS rates were 94.2% and 54.0% for clinical ≤T2 and ≥T3, respectively ($p < 0.001$; Figure 2A), 86.0% and 57.2% for patients without or with lymph node involvement, respectively ($p = 0.009$; Figure 1B), and 96.0% and 65.9% for tumor size <30 mm and ≥30 mm, respectively ($p < 0.001$; Figure 1C).

Using the Kaplan–Meier method, ECOG-PS 0, Alb levels ≥4.2 g/dL, Hb levels ≥13.2 g/dL, CRP levels <0.21 mg/dL, clinical T stage ≤2, clinically negative LN status, and tumor size <30 mm were significantly associated with better CSS; however, age, NLR, PLR, number of LNM, and pretreatment were not significantly associated with CSS. In the multivariate analysis, clinically positive LN status was a significant predictive factor for CSS in patients with PSCC (Table 2).

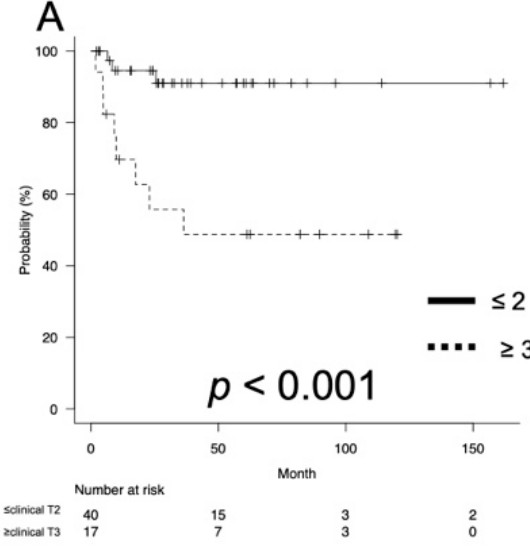

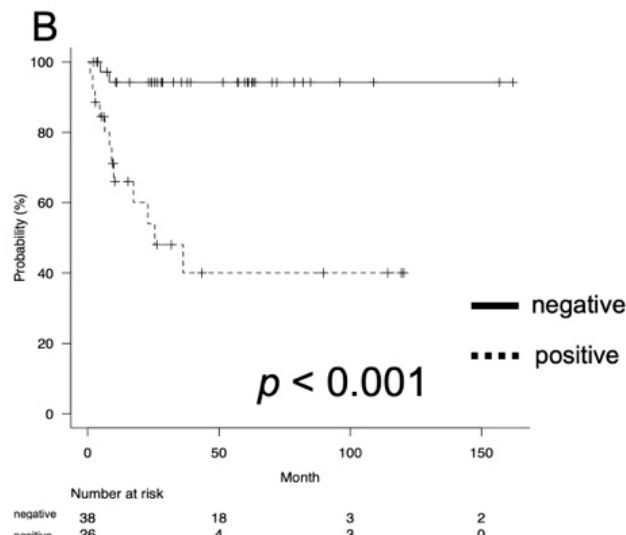

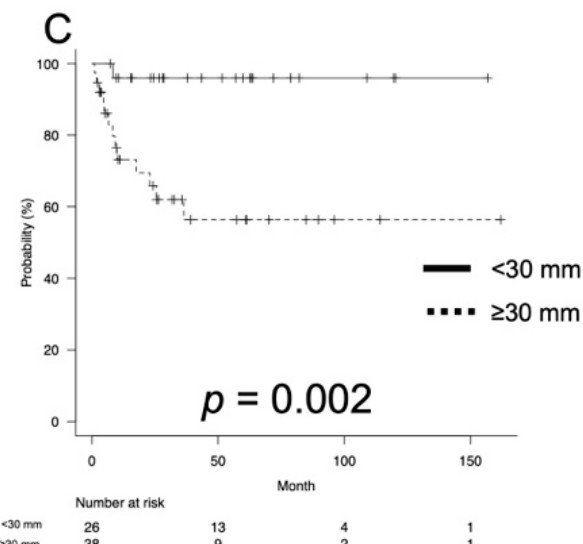

**Figure 2.** The Kaplan–Meier estimates of cancer-specific survival (CSS) according to the clinical T stage (**A**), lymph node status (**B**), and tumor size stratified by a cutoff of 30 mm (**C**). The 2-year CSS rates were 94.2% and 54.0% for clinical ≤T2 and ≥T3 stages, respectively ($p < 0.001$; Figure 2A); 86.0% and 57.2% for patients with or without lymph node involvement, respectively ($p = 0.009$; Figure 1B); and 96.0% and 65.9% for tumor size of <30 mm and ≥30 mm, respectively ($p < 0.001$; Figure 1C).

**Table 2.** Multivariate analysis.

| Variables | *p* Value | Odds Ratio | 95% Confidence Interval |
|---|---|---|---|
| Age (year) | | | |
| ≥74 | 0.967 | 1.027 | 0.298–3.537 |
| Tumor size (mm) | | | |
| ≥30 | 0.056 | 8.532 | 0.942–77.25 |
| Clinical T stage | | | |
| ≥3 | 0.348 | 0.526 | 0.137–2.014 |
| Clinical N stage | | | |
| ≥1 | 0.014 | 14.56 | 1.726–122.8 |
| C-reactive protein | | | |
| ≥0.21 | 0.080 | 3.536 | 0.861–14.52 |

## 4. Discussion

LNM was the strongest prognostic factor affecting oncological outcomes of patients with PSCC [3]. From the Memorial Sloan–Kettering Cancer Center surgical database between January 1995 and September 2011, the 3-year CSS rates of patients with positive, negative, and unknown LNM status were 90%, 65%, and 86%, respectively ($p = 0.03$) [18]. Previous studies of the prognosis of PSCC demonstrated that the histopathological characteristics of inguinal LNs, including the number of positive nodes, LN density, and PLNM, are predictive factors associated with CSS [8,18–22]. Based on the report from the Istituto Nazionale Tumori, the LN ratio (LNR) was an important prognostic parameter in a contemporary population of patients who underwent surgical treatment for PSCC and LNM [8]. An LNR threshold of 22% provided the most accurate discrimination of outcomes among patients with PSCC and LNM [8]. The 2- and 5-year CSS rates were 79.1% and 65.2% for patients with an LNR <22% and 35.9% and 9.6% for those with an LNR ≥22% ($p < 0.001$) [8]. Patients with PSCC with an LN density >20% had an increased risk of recurrence or progression in the univariate (hazard ratio (HR), 2.12; 95% confidence interval (CI), 1.18–3.80; $p = 0.011$) and multivariate (HR, 2.32; 95% CI, 1.21–4.45; $p = 0.011$) analyses [19]. Yu et al. reported that the CSS of patients with PSCC with an LNR >0.23% was significantly worse than that of those with an LNR ≤0.23% [20]. However, positive lymph node count (PLNC) was not associated with CSS in the multivariate analysis [20]. Thus, they concluded that LNR was a better prognostic factor than PLNC for PSCC [20]. Although a small number of patients underwent LN dissection and had their pathological LN status evaluated, we could not consider the pathological LN status since clinical LN involvement was determined to be a CSS predictor.

Conversely, the association between clinical T stage and prognosis for PSCC remains unclear. Moses et al. reported 5-year CSS rates for pTis, pT1, pT2, and pT3/4 of 100%, 84%, 54%, and 54%, respectively [23]. Based on the National Cancer Database, local excision (39%) and partial penectomy (38%) were most commonly performed for patients with PSCC [24]. Patients with clinical Tis/Ta or T1 disease were more often treated with non-penectomy ($p < 0.05$); cT2–T3 patients were more likely treated with penectomy ($p < 0.001$) [24]. No survival differences were observed between the penectomy (49.3 months) and non-penectomy approaches (50.3 months) in the overall cohort ($p = 0.107$) and when stratified by T stage ($p > 0.20$ for all) [24]. Indeed, primary operative therapy of PSCC is given the highest priority [10]. However, the proportion of partial resections in early clinical stages (pathological Tis, Ta, and T1) decreased from 60.2% to 51.7% between 2000 and 2018, although 94.4% of patients who had pathological T2–T4 disease underwent partial or total penectomy according to the Working Group of German Tumor Centers and the Society of Population-based Cancer Registries in Germany database [10]. Regarding primary tumor size, ≥3 cm was an independent prognostic factor for OS and CSS of PSCC [25]. However, the tumor size was not associated with CSS in this study. In the United Kingdom, a higher rate of patients with PC were also treated with organ-preserving surgery and improved LN management [26]. Thus, the treatment of LNM may be more important in patients with PSCC.

PLNM is a major prognostic factor in patients with PSCC, resulting in a 5-year survival rate of 12–33% [21,22]. Approximately one-third of patients with inguinal LNM (ILNM) from PCC have PLNM [22]. Previous studies reported that several factors for PSCC patients with ILNM, including the number of positive LN, the tumor grade of the involved nodes, the LNR, and the LN diameter, were histologically associated with PLNM [21,22]. According to CSS, PLNM (HR 2,07; $p = 0.007$) and bilateral LNM (HR 2.37; $p < 0.001$) were independent prognostic factors in the multivariate analysis, although PLND, bilateral LNM, ≥3 positive ILNM, and the LNM diameter were associated with CSS [23]. Zhao et al. reported that the LNM size, particularly >2 cm, was an independent predictor with an objective response rate and oncological outcomes in patients with esophageal SCC who did not undergo surgical treatments [27]. However, the clinical LN involvement in this

study was an independent significant cancer-specific survival (CSS) predictor, although the maximum LN diameter and number of LNM were not associated with CSS.

Here, we investigated the utility of inflammatory markers in identifying the prognosis of patients with PSCC. The association between cancer progression and inflammation is widely accepted, with tumors often presenting with characteristics of inflamed tissues, including immune cell infiltration or activated stroma [14,28]. Inflammation generates cancer-promoting microenvironmental changes and systemic changes that are favorable for cancer progression [14]. The systemic inflammatory response that is usually measured using surrogate peripheral blood-based variables independently predicts the clinical outcomes of various cancers [11–14,29–31]. Azizi et al. reported that pretreatment NLR is an independent predictor of overall survival [11] and suggested that patients with an elevated NLR ($\geq$3.0) were at an increased risk of pathological LNM according to the univariate analysis [11]. Likewise, patients with an NLR $\geq$2.8 or an LMR <3.3 had a significantly higher T stage ($p = 0.013$) and worse CSS ($p = 0.022$) than those with a high LMR [14]. Thus, NLR and LMR as inflammatory biomarkers may be useful for predicting prognosis in patients with PSCC [14]. Steffens et al. reported that a high preoperative serum CRP level was associated with poor survival in patients with PSCC [32]. In the univariate analysis, a high CRP level was a significantly poor prognostic factor for CSS. However, there were no significant differences between CSS and CRP levels in the multivariate analysis (CRP: odds ratio 3.54, 95% confidence interval 0.861–14.52).

Our study had some limitations. First, it was a retrospective study of multi-center data; thus, it was susceptible to potential bias owing to diagnostic and therapeutic differences among participating institutions. Second, the study had a relatively small sample size and a relatively short follow-up period. Therefore, a longer careful observation period of the oncological consequences is necessary. Third, we did not investigate the HPV infection rate. Additionally, the tumor grade, the anatomical infiltration level, and the tumor infiltration pattern were not assessed in this study. These points are critical limitations when analyzing the oncological outcomes in patients with PSCC. Therefore, careful interpretation of the results with predictive prognostic factors in patients with PSCC from this study may be necessary. Finally, the treatment was determined at the primary doctor's discretion and/or the patient's preference and general condition.

## 5. Conclusions

PSCC is a rare malignant neoplasm of genitourinary cancer, and cases of PSCC with LNM at the initial diagnosis have a relatively poor prognosis. Therefore, the treatment strategy for PSCC plays a very important role in improving oncological outcomes. In this study, the clinically positive LN status was a significant CSS predictor in patients with PSCC. Further prospective large-scale and long-term studies are required to validate our findings.

**Author Contributions:** M.K.: Protocol/project development, data collection and management, data analysis, manuscript writing/editing. K.T.: Data collection and management. K.K. (Kei Kawada): Data collection and management. T.I.: Data collection and management. M.T. (Masayuki Tomioka): Data collection and management. T.E.: Data collection and management. S.F.: Data collection and management. T.T.: Data collection and management. H.I.: Data collection and management. T.Y.: Data collection and management. K.K. (Kota Kawase): Data collection and management. D.K.: Data collection and management. M.T. (Manabu Takai): Data collection and management. K.I.: Data collection and management. K.N.: Data collection and management. T.K.: Protocol/project development, data management, manuscript writing/editing. All authors have read and agreed to the published version of the manuscript.

**Funding:** This research received no external funding.

**Institutional Review Board Statement:** The study protocol was approved by the Institutional Review Board of Gifu University (number: 2020-271). All procedures performed in studies involving human participants were in accordance with the ethical standards of the institutional and/or national research committee and with the 1964 Helsinki declaration and its later amendments or comparable ethical standards.

**Informed Consent Statement:** For this type of study formal consent is not required. Pursuant to the provisions of the ethics committee and the ethic guideline in Japan, written consent was not required in exchange for public disclosure of study information in the case of retrospective and/or observational study using a material such as the existing documentation. The study information was open for the public consumption at http://www.med.gifu-u.ac.jp/file/2020-271.pdf (accessed on 21 November 2021).

**Data Availability Statement:** The data presented in this study are available on request from the corresponding author. The data are not publicly available due to privacy and ethical reasons.

**Conflicts of Interest:** The authors declare no conflict of interest.

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
