# Peer review of "Clinical Lymph Node Involvement as a Predictor for Cancer-Specific Survival in Patients with Penile Squamous Cell Cancer"

_curroncol, doi:10.3390/curroncol29080432_

Round 1
Reviewer 1 Report
Quality of the manuscript is improved after revision
Reviewer 2 Report
All concerns have been addressed. The manuscript is now acceptable in its present form.
This manuscript is a resubmission of an earlier submission. The following is a list of the peer review reports and author responses from that submission.
Round 1
Reviewer 1 Report
Authors studied a fair number of patients treated surgically for penile cancer with long term follow-up. Using simple laboratory, clinical and pathological data, they found good predictors of survival. The main factor would be the size of lymph node metastasis. The only comment I would made would be the choice of the size of metastasis. There are only 6 patients which metastasis is larger than 40 mm. How is the graphic if the median size (18 mm) is considered? I believe this information is important; otherwise, it could seems like a statistical manipulation.
Reviewer 2 Report
In this study, the authors aim to assess the prognostic role of nodal size and other variables in patients with penile SCC. Some issues need to be addressed:
- it is not clear whether the assessed lymph nodes are metastatic or not. The authors should clearly state that we are dealing here with the size of metastatic nodes.
- the prognostic role of nodal size has been already reported in SCCs of the head and neck, the authors should cite and comment these articles, such as 10.1111/ans.16413, 10.1007/s00268-016-3675-y, 10.3389/fonc.2020.00523
- information about histopathology patterns and grade is lacking, which would provide important prognostic information; this is a fundamental limitation, along with the absence of information about HPV status.
- line 149: it's Istituto Nazionale Tumori, please check it out
Reviewer 3 Report
The manuscript by Kawase et.al. explored association between certain inflammatory markers and tumor/LN clinical characteristics with survival outcome in a small cohort of 64 patients with penile cancer. They made the conclusion that the size of LN > 40 mm predicts poorly cancer specific survival. While this conclusion is consistent with the data presented, it adds little value to our current understanding of penile cancer outcome.
Major concerns
1. The authors used maximum diameter of LN size as a predictor for CSS with the cut off of 40 mm. LN of this size is not commonly seen clinically - looking at the authors data, only 6 patients (9%) of the entire cohort had LN with size above this cut off - which means 40 mm is towards the extreme end of the size distribution. By using this cut off, there is significant selection bias involved because only patients with the most advanced disease in LN were included. As such, it was not surprising these patients exhibited worse outcome. An alternative is to treat LN size as a continuous variable which makes better sense statistically.
2. The size of LN is not included as a criterion in the widely accepted TNM staging of penile cancer, where most of the outcome study is based on. While I applaud the initiative from authors to explore the utility of size of LN in outcome prediction, they should compare it with traditional tumor N staging (e.g. N0-3) as a predictor and determine which one is better at risk prediction.
3. The authors showed elevation of inflammatory markers is associated with worse survival - this is not a new finding and has been reported multiple times. Additional literatures on the association between CRP and poorer survival should be included. e.g. Steffens et.al. BMC Cancer 2013 etc.
Minor concerns
1. Table 1, clinical N stage is categorized as negative and positive. Does this refer to non-palpable vs palpable?
2. When were labs checked? At the time of diagnosis? Surgery?
3. It remains unclear how the cut off values for covariates were determined (line 94-95). This was not explained in reference 16. Please clarify.
4. Line 126, "Alb" should be "Hb".
5. Line 158, please define "PLNC".
6. Line 203 "no significant differences between CSS and inflammatory biomarkers in multivariate analysis" - need to include the OR with CI in the results.